# Human Papillomavirus Self-Sampling for Primary Cervical Cancer Screening in Under-Screened Women in Hong Kong during the COVID-19 Pandemic

**DOI:** 10.3390/ijerph19052610

**Published:** 2022-02-24

**Authors:** Siew-Fei Ngu, Lesley S. K. Lau, Justin Li, Grace C. Y. Wong, Annie N. Y. Cheung, Hextan Y. S. Ngan, Karen K. L. Chan

**Affiliations:** 1Department of Obstetrics and Gynaecology, The University of Hong Kong, Queen Mary Hospital, Hong Kong; ngusiewf@hku.hk (S.-F.N.); lsk382@hku.hk (L.S.K.L.); hysngan@hku.hk (H.Y.S.N.); 2Li Ka Shing Faculty of Medicine, The University of Hong Kong, Hong Kong; just.li.328@gmail.com; 3The Family Planning Association of Hong Kong, Hong Kong; wongcyg@yahoo.com; 4Department of Pathology, The University of Hong Kong, Queen Mary Hospital, Hong Kong; anycheun@pathology.hku.hk

**Keywords:** human papillomavirus (HPV), self-sampling, cervical cancer screening, COVID-19 pandemic

## Abstract

The aim of this study was to assess the effectiveness of HPV self-sampling for cervical cancer screening and the best means of service delivery, with a specific focus on under-screened women, particularly during the COVID-19 pandemic. Using three arms of service delivery (social media, school outreach and underserved outreach), we recruited under-screened women aged 30–65 years from two population groups: the general public and specific underserved communities, from whom self-sampled specimens and optional clinician-sampled cervical specimens were obtained for HPV testing. A total of 521 self-sampling kits were distributed, of which 321 were returned, giving an overall uptake rate of 61.6%. The response rate was higher in the face-to-face underserved outreach (65.5%) compared to social media (22.8%) and school outreach (18.2%). The concordance for HPV detection between self-sampled and clinician-sampled specimens was 90.2% [95% confidence interval (CI) 85.1–93.8%; Cohen’s kappa 0.59 (95% CI 0.42–0.75)]. Overall, 89.2% of women were willing to have self-sampling again. In conclusion, HPV self-sampling is an effective method for cervical cancer screening and can be considered as an option, particularly in women who are reluctant or unable to attend regular screening. Various service deliveries could be considered to increase participation in cervical cancer screening.

## 1. Introduction

Cervical cancer is the fourth most common cancer in women worldwide, with more than 604,000 women diagnosed and around 341,000 women dying from the disease in 2020 alone [1]. In Hong Kong, it is the eighth most common cancer in women and ranked eighth in overall cancer mortality in 2019 [2]. Despite the launch of population-based cervical screening in 2004 by the Department of Health, the incidence rate of cervical cancer has remained relatively static following an initial decline observed immediately after the programme’s initiation. Furthermore, a recent Health Behaviour Survey conducted in Hong Kong in 2018/19 found that only 45.8% of women aged 25–64 had ever had a cervical smear test without suspecting symptoms of cervical cancer prior to the test [3]. This reported rate of cervical screening coverage is relatively low compared to the 70–80% achieved in other developed countries [4]. The reasons for non-attendance are likely to be multifactorial, including a lack of time, inconvenience, embarrassment, pain, discomfort, cultural objections, transportation and cost [5]. This is detrimental, as more than half of invasive cervical cancers are diagnosed in never- or under-screened women. Therefore, strategies to increase the uptake of cervical screening is important in reducing cervical cancer incidence and mortality.

In recent years, high-risk human papillomavirus (hrHPV)-based primary cervical cancer screening has been introduced in several regions, including Hong Kong. It has been found to be more sensitive than conventional cytology in the detection of high-grade cervical intraepithelial neoplasia (CIN2+) [6], with a better negative predictive value and longer screening interval [7]. With the availability of hrHPV-based screening, the option of self-sampling, where a vaginal sample is self-taken by a woman for hrHPV testing, offers an alternative screening strategy and an opportunity to improve screening coverage [8]. Self-sampling may be able to mitigate some barriers to the non-attenders mentioned above, as the sample can be obtained in the privacy of a woman’s own home. A recent meta-analysis reported that both the sensitivity and specificity of hrHPV testing on self-samples were similar to that of clinician-samples, particularly when validated PCR-based HPV tests were used [9].

The COVID-19 pandemic has resulted in worldwide disruptions of health care in general, including cervical cancer screening services [10]. The disruption may jeopardise the WHO’s global strategy to expedite the elimination of cervical cancer as a public health concern. The screening and prompt treatment of women diagnosed with cervical cancer are two out of the three targets of this initiative. Even with the resumption of screening activity after the COVID-19 outbreaks were controlled, many women are reluctant to access the service due to fears of being exposed to SARS-CoV-2 on the way to, or at, the healthcare centre. Consequently, HPV self-sampling has the advantage that it can be performed at home without attending health care settings.

In order to incorporate HPV self-sampling to local screening programmes, this study aimed to assess the effectiveness of self-sampling and the best means of service delivery, with a specific focus on women who were reluctant or unable to attend regular cervical cancer screenings.

## 2. Materials and Methods

### 2.1. Trial Design

This study was approved by the institution’s ethics review board and conducted in accordance with the principles of the Declaration of Helsinki. Using three arms of service delivery, we recruited 521 women from two population groups: the general public and specific underserved/underprivileged communities, from whom we obtained self-sampled specimens for hrHPV testing and optional clinician-sampled cervical specimens for cytological assessment and hrHPV testing.

### 2.2. Study Population

Women aged 30–65 years with a history of sexual activity who were never or under-screened (no cervical screening in the past 3 years) were eligible for the study. Those with a history of hysterectomy, currently pregnant, undergoing treatment for cervical intraepithelial neoplasia or cancer, history of HPV vaccination or symptoms of cervical cancer were excluded. Women were recruited from three arms of service delivery: ‘General Public Through Social Media’, ‘School Outreach Programme’, and ‘Underserved Outreach through non-governmental organisations (NGOs)’.

#### 2.2.1. General Public through Social Media

A social media campaign designed to raise awareness of cervical cancer and HPV, while introducing self-sampling as a new potential screening method through a standardised educational and an instructional video, was launched on the NGO’s Facebook page and website. Viewers were redirected to a ‘plug-and-play’ multi-functional platform that served to educate visitors on HPV and allowed participants to register for the study. Those who signed up to the platform and consented to participate in the study received a self-sampling kit by post on a first-come, first-served basis until 167 kits allocated to this group had been sent out.

#### 2.2.2. School Outreach Programme

We partnered with an NGO, whose existing ‘Mother-Daughter’ programme had provided the HPV vaccine to local schoolgirls. An E-notice was initially sent to the mothers whose daughter were in the HPV vaccination programme, and later to the whole school. A website link to the standardised educational and instructional video, and information about the study was included in the E-notice. Those who consented to participation received a HPV self-sampling kit by post on a first-come, first-served basis until 167 kits allocated to this group had been sent out.

#### 2.2.3. Underserved Outreach through NGOs

We partnered with two NGOs who are familiar with the targeted underserved demographic, namely the ethnic minority and ethnic Chinese populations, to recruit participants to pre-scheduled health-talks on cervical cancer and its screening, with a specific focus on HPV and self-sampling as a new potential primary screening tool. The health-talks were delivered either face-to-face or online. A short standardised instructional video on the self-sampling procedure was played after the talk. Following this, all participants (*n* = 187) received a HPV self-sampling kit, either in person (if attended face-to-face talks) or by post (if attended online talks).

### 2.3. Study Intervention

Women who were interested in participating were given information about the study and were requested to indicate whether they would like to participate. Those who consented to participation were given a self-sampling kit and also invited for a clinician-collected cervical smear for co-testing. The self-sampling kit included a sealed long sterile Dacron swab (FLOQSwab #552C by COPAN Diagnostics, Brescia, Italy) with a plastic sleeve, a plastic bag complete with the participant’s identification number, a set of diagrammed instructions for self-sampling and an acceptability questionnaire. Women were instructed to insert the swab 5 cm into the vagina, rotate it fully for 10 s, and return the swab to their plastic sleeve. They were instructed to collect the self-sample at their convenience in a place of their choosing. Having done this, they were requested to return the self-collected samples and the completed questionnaire to a pre-designated FPA clinic, or by post if they opted for self-sampling only.

In order to obtain a comparative clinician-sampled specimen, all participants were offered a smear-taking appointment upon delivery of their self-sampled specimens at an FPA clinic of their choice. A cervical smear was taken by a qualified smear-taker using a Cervex-brush and put into a liquid medium. Clinician-collected specimens underwent cytology assessment and HPV testing (co-test), while self-collected specimens underwent HPV testing only. In the laboratory, the self-collected dry swabs were resuspended in a labelled sterile secondary tube containing 4 mL of PreservCyt media (Hologic, Marlborough, MA, USA) [11]. All specimens were tested for hrHPV using the cobas^®^4800 HPV test (Roche Molecular Diagnostics, Pleasanton, CA, USA), which is an FDA approved test for hrHPV testing. This is a fully automated real-time PCR targeting the viral L1 region and simultaneously detecting 14 hrHPV types. The test specifically identifies HPV types 16 and 18, while concurrently detecting 12 other HPV types (31, 33, 35, 39, 45, 51, 52, 56, 58, 59, 66 and 68). For all specimens, 1 mL of aliquot was loaded on the cobas instrument, and all procedures were performed according to the manufacturers’ instructions.

During the study period, 521 self-sampling kits were sent out to participants who consented. Of these, 208 women agreed to both self and clinician-sampling, 113 women submitted self-sampling alone, and 2 women had clinician-sampling alone. Abnormal results were managed according to the Hong Kong College of Obstetricians and Gynaecologists guidelines for cervical cancer prevention and screening [12]. Participants who undertook both self- and clinician-sampling were informed of their clinician-sampled results only, as this is currently the gold standard. Those who opted for self-sampling only were informed of their results along with a disclaimer explaining that their results should not be considered diagnostic and advising them to consult a clinician for a conventional cervical smear.

### 2.4. Study Outcome and Statistical Analysis

The primary outcome was the uptake rate of HPV self-sampling (the overall rate, as well as the individual rate of each recruitment method), calculated by the number of self-samples returned/number self-samples kit sent. The secondary outcomes were the feasibility (response rate, calculated by the number of participants registered/number of participants approached) of the different community approaches to service delivery, the attitudes of women towards HPV self-sampling and the concordance between the self-sampled and clinician-sampled results (assessed using the Kappa statistic). Sociodemographic data and previous experience of cervical smears were collected. The acceptability of self-sampling was assessed using a five-item Likert scale that covered a range of subjective qualities such as convenience, embarrassment, confidence, discomfort and overall experience. The results of the pathological assessment and questionnaire responses were entered into an electronic database. The data entry was double checked for inconsistency before analysis. The difference in the proportions amongst the different groups were calculated using a proportion test in R with a significance level of 0.05.

## 3. Results

### 3.1. Demographics and Response Rate

The demographic data were available for 316 participants who returned the questionnaire and are shown in Table 1. The median age was 43 years. Of these, 119 (37.7%) women had never had a cervical cancer screening.

#### 3.1.1. General Public through Social Media

A media campaign on HPV and self-sampling was shown 16,300 times to 14,256 users on the NGO’s Facebook page during a one-week period (8–14 September 2020). It received 1247 clicks (click-through rate of 6.69%) and delivered 98 conversions (participants who registered and consented to participation) during the one-week period. An additional of 186 participants registered subsequently outside the media campaign period. The response rate (calculated by number of participants registered/number of clicks received) for this service delivery was 22.8% (Table 2). The participants in this group comprised mostly of Filipino domestic helpers (42.0%) and Chinese women (30.9%).

#### 3.1.2. School Outreach Programme

An E-notice was initially sent out to 514 mothers whose daughters were in the HPV vaccination program, of which 94 (18.2%) registered for the study. The e-notice was then sent to the whole school (including mothers who were in the HPV vaccination program), and 81 (6.8%) registered (Table 2). Of note, most women in this group were Chinese in the 41–50 age group.

#### 3.1.3. Underserved Outreach through NGOs

During the COVID-19 pandemic, health-talks were delivered online to 177 underserved women, of which 14 (7.9%) registered for the study. When the pandemic situation improved, health talks were conducted face-to-face for 264 participants, where 173 (65.5%) registered (Table 2). The underserved women in this group consisted of two distinct populations, the Filipino domestic helpers (56.2%) and the Chinese immigrants who had recently moved to Hong Kong (43.8%).

### 3.2. Uptake Rate

A total of 521 self-sampling kits were distributed to the three arms of service delivery, of which 321 self-samples were returned, giving an overall uptake of HPV self-sampling of 61.6%. The uptake rates of HPV self-sampling were 48.5%, 47.9% and 86.6% for the ‘General Public Through Social Media’, ‘School Outreach Programme’ and ‘Underserved Outreach through NGOs’, respectively (Table 2). Two women in the ‘School Outreach Programme’ attended for clinician-collected samples for co-testing but did not return a self-sample. Of the 521 participants who were sent the self-sampling kit, 210 (40.3%) had clinician-collected samples for co-testing. The uptake rates of clinician-sampling for the ‘General Public Through Social Media’, ‘School Outreach Programme’ and ‘Underserved Outreach through NGOs’ were 34.7%, 31.7% and 52.9%, respectively. Out of the 321 participants who returned the self-samples, 208 (64.8%) had both self-samples and clinician-samples, while 113 (35.2%) (28.4% in ‘General Public Through Social Media’, 33.8% in ‘School Outreach Programme’ and 38.9% in ‘Underserved Outreach through NGOs) submitted self-samples alone.

### 3.3. HPV Test/Co-Test Results

A total of 200 women had self-samples only, 121 women had both self-samples and clinician-samples, while 2 women had clinician-samples only. Among the self-sampled specimens, hrHPV was detected in 35 (10.9%) women, where 4 (11.4%) were HPV 16/18, and 31 (88.6%) were other hrHPV types (Table 3a). After the participants received their self-sampled results and were advised to have a clinician-collected sample, a further 87 women attended FPA for a conventional cervical smear for co-testing. The median time interval between the two samples was 1.5 months (range 0.6–3.8 months). Among 35 women with hrHPV detected on self-samples, 29 (82.9%) attended for follow-up clinician-collected cervical smear for cytology. Overall, 210 (65.4%) participants had clinician-collected samples for co-testing and the results are shown in Table 3b. hrHPV was detected in 27 (12.9%) women, where 1 (3.7%) was HPV 16/18, and 26 (96.3%) were other hrHPV types. Fourteen women (6.7%) had abnormal cytology and were referred for colposcopy. Of these, 13 women had a positive hrHPV test. Colposcopic guided cervical biopsy found that one had high-grade squamous intraepithelial lesion (SIL), seven had low-grade SIL, four had normal histology, and information was not available for two women. The concordance (overall rates of agreement) for hrHPV detection between self-sampled specimens and clinician-sampled specimens were 90.2% (95% confidence interval (CI) 85.1–93.8%), with Cohen’s kappa of 0.59 (95% CI 0.42–0.75). The negative percent agreement and positive percent agreement were 94.9% (95% CI 90.2–97.5%) and 62.1% (95% CI 42.4–78.7%), respectively.

### 3.4. Acceptability

Of the 321 women who returned the self-samples, 316 (98.4%) women completed the acceptability questionnaire. Of these, 71.1% women found self-sampling convenient/very convenient, and 69.2% were not embarrassed/not embarrassed at all with self-sampling (Table 4). In terms of acceptability in the different social groups, a higher proportion of participants found self-sampling easy/very easy and were confident/very confident with self-sampling in the social media group (60.3% and 59.0%, respectively) and among foreign domestic helpers in the underserved outreach group (67.1% and 71.3%, respectively) than in the school outreach group (46.1% and 46.2%, respectively) or among new Chinese immigrants in the underserved outreach group (52.9% and 49.3%, respectively). However, a significantly lower proportion of participants recruited from the school outreach programme had an overall good or very good experience with self-sampling compared to the social media group (30.3% vs. 53.2%, *p* = 0.007). Regarding the different age groups, a significantly lower proportion of women in the older age group of 51–65 were confident/very confident with self-sampling, compared to the younger age group of 30–40 (34.2% vs. 64.0%, *p* = 0.002). The acceptability of self-sampling was similar in terms of previous smear experience, parity and education background. 

Overall, 263 (89.2%) women were willing to have self-sampling again because most found it simple to use, while 32 (10.8%) women were not willing because they were either not confident or preferred healthcare to take the samples (Table 5). The most preferred cervical cancer screening method was self-sampling (*n* = 97, 32.8%), followed by clinician-collected cervical smear for co-test (*n* = 76, 25.7%), and clinician-collected vaginal swab for HPV test (*n* = 41, 13.8%), while 82 (27.7%) women had no preference on the screening methods.

## 4. Discussion

Most women who are diagnosed with cervical cancer have never been screened or do not engage in regular cervical cancer screening [9]. Yet, recent studies indicate an increasing burden of cervical cancer, even in regions with good screening coverage and well organised cytology-based programmes [13,14]. As demonstrated in the most recent population-based health survey, a significant proportion (57%) of women in Hong Kong have never been screened or are under-screened, making them susceptible to cervical cancer [3]. Furthermore, because of the COVID-19 pandemic, many women defer cervical cancer screening due to a reluctance to attend healthcare centres. Therefore, there is an urgent need to increase the screening coverage, as well as improve the effectiveness of the recruitment method in reaching the never or under-screened population.

Of the three recruitment approaches explored in this study, the face-to-face community campaign for underserved women yielded a higher response rate (65.5%) and uptake rate (86.6%) compared to the social media and school campaigns. One of the reasons may be due to the direct offer of the self-sampling kit to the women after the face-to-face health talks, as opposed to delivery by mail in the social media and school campaigns. In a recent meta-analysis, door-to-door campaigns where the self-sampling kits were delivered by community health workers to the women’s homes or workplaces was shown to be more effective (pooled participation rate 94.2%; 95% CI 80.2–100%) in reaching under-screened populations compared to when the women had to request a self-sampling kit (pooled participation rate 7.8%; 95% CI 5.2–10.9%) [9]. Our results are comparable to other studies included in the meta-analysis, demonstrating that face-to-face recruitment is an effective way to engage women to participate in cervical cancer screening [9]. Considering the requirements for social distancing during the COVID-19 pandemic, and the premise that women do not want to attend clinics to avoid potential exposure, it seems counter intuitive that the highest response rate was in the women who were approached in the face-to-face community campaign. This is likely because the face-to-face health talks were conducted in the community centres when there were very few local cases of COVID-19. However, direct offers of the self-sampling kits in the face-to-face community campaign is limited to specific populations visiting specific centres, and not appropriate for the general population. This strategy also requires more resources such as manpower and facilities and, thus, is potentially less cost effective, making social media outreach more appealing. Digital outreach through social media has the potential to reach out to more women and would be beneficial during the COVID-19 pandemic due to social distancing rules that prevent the face-to-face delivery of health talks. However, this approach is not appropriate for reaching a rural population, but rather an urban one. 

The concordance in our study (kappa 0.59) was lower than that of the systemic review by Petiginat et al. (kappa 0.66) [15] and a study by Tranberg et al. (kappa 0.70) [16]. This difference may be due to the variation in the self-sampling kits, HPV tests and study populations. In our study, some of the inconsistency in the HPV concordance between the self-samples and clinician-samples could be because a significant proportion of the participants only had clinician-collected samples after receiving their self-sampled results. The time interval between the two samples could predispose them to a new HPV infection or spontaneous clearance of a previous infection. We observed that the prevalence of hrHPV was slightly higher in the clinician-collected samples (12.9%) compared with the self-samples (10.9%), but the trend in prevalence of HPV 16/18 and other hrHPV types was comparable.

Self-sampling is an acceptable alternative to clinician-sampling and can potentially increase the opportunity to detect women who are at risk of developing cervical cancer. The convenience of service delivery by self-sampling may encourage participation in the screening programme [8]. We found that 89.2% of the under-screened women in our study were willing to have self-sampling again, as it can be asserted that obtaining a sample in the privacy of one’s own home may eliminate some of the factors that currently deter women from attending cervical screenings. Self-sampling is recognised as an easy, convenient collection method, with most participants being confident with the procedure and having a positive experience and minor discomfort, as found in several studies [17,18,19] including ours. Nonetheless, the acceptability of self-sampling is highly variable among different settings, social backgrounds and age groups [20,21]. In our study, women in the older age group were significantly less confident in performing self-sampling compared to the younger age group, which contrasts with another local study where the confidence was similar among younger and older age groups [17]. One potential reason is that the women in the older age group may have experienced conventional smears previously and feel more confident with clinician sampling. Some of the disadvantages of self-sampling included the possibility of inaccurately performing the test, hurting oneself during the process and the accuracy of the test, which could affect the results of the study. These may be some of the potential reasons why 38.4% of the participants in our study who were sent the self-sampling kit did not return the self-samples. Consequently, apart from more education on the HPV self-sampling procedure, a better sampling kit is needed. For example, a sampling kit that has an indicator that the sample is satisfactory, such as a change of colour on the collection medium or a control indicator, could help to provide better reassurance and confidence in handling the procedure.

In our study, a significant proportion of participants recruited through social media were domestic helpers from the Philippines, which may have led to a skewed result. Nevertheless, it proves that social media is a good platform to connect with domestic helpers. In 2020, there were more than 373,000 foreign domestic workers in Hong Kong, with around 207,000 from the Philippines, mostly women [22]. Many of these women are less likely to be screened, are unfamiliar with the local healthcare settings and, thus, only seek medical advice when they are symptomatic [23]. Identifying and engaging these women in cervical cancer screening will be challenging. Hence, our findings shed some light on effective communication strategies for this population. Another limitation of our study is the small sample sizes in each group of service delivery and, thus, these samples may not be representative of the larger local and immigrant minority populations. Future research focusing on the feasibility and acceptability of HPV self-sampling should focus on increasing the numbers of participants to include a diverse background, whilst appraising recommendations made in previous studies to enhance the uptake of cervical cancer screening.

## 5. Conclusions

In conclusion, HPV self-sampling is an effective method for cervical cancer screening in Hong Kong and can be considered as an option, particularly in women who are reluctant or unable to attend regular screening. Various service deliveries, including face-to-face community outreach, social media and school outreach, could be considered to increase participation in cervical cancer screening.

## Figures and Tables

**Table 1 ijerph-19-02610-t001:** Demographics.

Demographics	General Public through Social Media, *n* (%)	School Outreach Programme, *n* (%)	Underserved Outreach through NGOs, *n* (%)
No. of patients	81	78	162
Age			
30–40	45 (55.6)	8 (10.0)	76 (46.9)
41–50	25 (30.9)	56 (70.0)	70 (43.2)
51–65	11 (13.6)	16 (20.0)	16 (9.9)
Ethnicity			
Chinese	25 (30.9)	72 (90.0)	71 (43.8)
Filipino	34 (42.0)	0	91 (56.2)
Asian (not-specified)	11 (13.6)	3 (3.8)	0
Unknown	11 (13.6)	5 (6.3)	0
Returned Questionnaire	81 (100.0)	78 (100.0)	157 (96.9)
Monthly income (HKD)			
<$10,000	39 (48.1)	3 (3.8)	81 (51.6)
$10,000–$19,999	13 (16.0)	13 (16.7)	29 (18.5)
$20,000–$29,999	5 (6.2)	11 (14.1)	15 (9.6)
$30,000–$39,999	5 (6.2)	14 (17.9)	5 (3.2)
≥$40,000	7 (8.6)	26 (33.3)	0
Missing	12 (14.8)	11 (14.1)	27 (17.1)
Education			
Primary or below	1 (1.2)	1 (1.3)	13 (8.3)
Secondary	34 (42.0)	45 (57.7)	80 (51.0)
Tertiary	29 (35.8)	25 (32.1)	33 (21.0)
Postgraduate or above	15 (18.5)	5 (6.4)	21 (13.4)
Missing	2 (2.5)	2 (2.6)	10 (6.3)
Parity			
0	20 (24.7)	0	12 (7.6)
≥1	54 (66.7)	69 (88.5)	136 (86.7)
Missing	7 (8.6)	9 (11.5)	9 (5.7)
Last smear *			
Never	29 (37.7)	14 (18.4)	76 (48.4)
>3 years ago	20 (26.0)	27 (35.5)	31 (19.7)
>5 years ago	22 (28.5)	18 (23.6)	34 (21.6)
Missing	6 (7.8)	17 (22.4)	16 (10.2)

* 6 participants mentioned that they had had their last smear within 3 years in the questionnaire.

**Table 2 ijerph-19-02610-t002:** Participation and uptake rate of HPV self-sampling.

Group	Method	Number of Participants Approached	Number of Participants Registered	Response Rate	Kit Sent	Kit Returned/ Co-Test Done	Uptake Rate
General Public Through Social Media	Facebook Advertisements	Ads showed 16.3 K times to 14,256 users. It received 1247 clicks (click-through rate: 6.69%). It delivered 98 conversions from 8–14 September 2020. 186 registered outside this period.	284	22.8%	167	81	48.5%
School Outreach Programme	E-notice to mothers in the HPV vaccination program	514	94	18.2%	167	80 (2 had co-test only)	47.9%
	E-notice to the whole schools *	1190	81	6.8%			
Underserved Outreach through NGOs	Face-to-face	264	173	65.5%	187	162	86.6%
	Online	177	14	7.9%			

* Including 409 parents who were in the vaccination program.

**Table 3 ijerph-19-02610-t003:** (**a**) Self-sampling HPV results. (**b**) Co-test results (N = 210).

**(a)**				
**Group**	**N**	**HPV Negative, *n* (%)**	**HPV 16/18** **Positive, *n* (%)**	**Other hrHPV** **Positive, *n* (%)**
General Public Through Social Media	80	67 (83.8)	0 (0)	13 (16.3)
School Outreach Programme	77	73 (94.8)	0 (0)	4 (5.2)
Underserved Outreach through NGOs	161	143 (88.8)	4 (2.5)	14 (8.7)
Total	318	283 (89.0)	4 (1.3)	31 (9.7)
Excluding 3 invalid results (1 in each group)				
**(b)**				
**Cytology**		**HPV Negative**	**HPV 16/18** **Positive**	**Other hrHPV** **Positive**
Negative		171	0	14
LSIL		1	0	2
ASCUS		11	1	8
ASC-H		0	0	2
HSIL		0	0	0

LSIL, low-grade squamous intraepithelial lesion; ASCUS, atypical squamous cell of undetermined significance; ASC-H, atypical squamous cell, cannot rule out high-grade lesion; HSIL, high-grade squamous intraepithelial lesion.

**Table 4 ijerph-19-02610-t004:** Acceptability for all women who had self-sampling (N = 316).

Attributes	*n* (%)	N
Easy or very easy	173 (56.9)	304
Convenient or very convenient	217 (71.1)	305
Not embarrassed or not embarrassed at all	213 (69.2)	308
Confident or very confident	173 (56.7)	305
No discomfort or no discomfort at all	153 (50.3)	304
Overall good or very good experience	133 (44.2)	301

N = total number of women who provided responses to the different attributes of the questionnaire (some women returned an incomplete questionnaire).

**Table 5 ijerph-19-02610-t005:** Women’s willingness to perform self-sampling again and reasons.

Willingness	*n*	%
Yes	263	89.2
Simple	167	63.5
Comfortable	95	36.1
Quick	92	35.0
Less embarrassed	85	32.3
Non-painful	74	28.1
Confident	50	19.0
No	32	10.8
Not Confident	16	50.0
Prefer healthcare-collected specimen	16	50.0
Hurt	8	25.0
Not easy	6	18.8
Painful	5	15.6
Not comfortable	4	12.5

## Data Availability

Not applicable.

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
