# Peer review of "Human Papillomavirus Self-Sampling for Primary Cervical Cancer Screening in Under-Screened Women in Hong Kong during the COVID-19 Pandemic"

_ijerph, 2022, doi:10.3390/ijerph19052610_

Round 1
Reviewer 1 Report
Please find some remarks and proposed modifications in the enclosed document

Reviewer 2 Report
|
In this manuscript, the authors performed a study to evaluate if self-sampling for primary cervical cancer screening would be effective in women belonging to under-represented segments in era of covid-19 pandemic. Based on their findings, they concluded that HPV self-sampling is an effective method for cervical cancer screening. There are some shortcomings that need to be addressed which are mentioned below: 1. The authors need to address the disadvantages of self-sampling in the discussion section. There is always a possibility of inaccurately performing the test, injuring themselves during the process, and the sensitivity of the test itself, which could affect the results of the study. 2. Under the study intervention section, it is unclear as to how many patients agreed/ consented for the clinical sample submission and the self-test and how many patients only provided the self-test sample alone. Please provide more clarification. 3. The authors are requested to pay attention to the table formatting. For example in table 2, the column headings are missing on page 6. 4. For table 4, it is stated that N=316 provided responses for the acceptability questionnaire; however, looking at different attributes of the questionnaire the N varies. Can the authors please explain this variability in the responses? 5. It seems counter intuitive that highest response rate was in the women who were approached in a face to face community campaign given the requirements for social distancing during the pandemic and the premise that women do not want to go to the clinics to avoid potential exposure. 6. Also, the success of social media outreach programs would also be impacted by the access to social media in the ‘underserved women’ and also level of education. Did the authors find any correlation between the education status and willingness to administer these self-tests?
|
